# Does an Alternative Sunitinib Dosing Schedule Really Improve Survival Outcomes Over a Conventional Dosing Schedule in Patients with Metastatic Renal Cell Carcinoma? An Updated Systematic Review and Meta-Analysis

**DOI:** 10.3390/cancers11121830

**Published:** 2019-11-21

**Authors:** Doo Yong Chung, Dong Hyuk Kang, Jong Won Kim, Do Kyung Kim, Joo Yong Lee, Chang Hee Hong, Kang Su Cho

**Affiliations:** 1Department of Urology, Inha University School of Medicine, Incheon 22212, Korea; wjdendyd@gmail.com (D.Y.C.); dhkang0424@inhauh.com (D.H.K.); 2Department of Urology, Urological Science Institute, Yonsei University College of Medicine, Seoul 06273, Korea; 3Department of Urology, Gangnam Severance Hospital, Yonsei University College of Medicine, Seoul 06273, Korea; doctor2play@yuhs.ac (J.W.K.); chhong52@yuhs.ac (C.H.H.); 4Department of Urology, Soonchunhyang University Seoul Hospital, Soonchunhyang University Medical College, Seoul 06273, Korea; dokyung80@hotmail.com; 5Department of Urology, Severance Hospital, Yonsei University College of Medicine, Seoul 06273, Korea; joouro@yuhs.ac

**Keywords:** renal cell carcinoma, sunitinib, alternative dosing, survival outcomes, adverse events, systematic review, meta-analysis

## Abstract

Treatment-related adverse events (AEs) can obfuscate the maintenance of a conventional schedule of sunitinib in patients with metastatic renal cell carcinoma. Accordingly, alternative schedules seeking to improve the safety profile of sunitinib have been tested. Recently, two meta-analyses similarly described improved safety profiles favoring a two weeks on and one week off (2/1) schedule, but with conflicting results for survival outcomes. Therefore, we conducted an updated systematic review and meta-analysis, including all recently published studies and using complementary statistical methods. Endpoints included progression-free survival, overall survival, and AEs of 15 types. Eleven articles were included in this meta-analysis. Using adjusted findings, we noted statistically better results in progression-free survival (hazard ratio, 0.58; 95% confidence interval, 0.39–0.84; *p* = 0.005), but no difference in overall survival (hazard ratio, 0.66; 95% confidence interval, 0.42–1.04; *p* = 0.08). Moreover, the 2/1 schedule was beneficial for reducing the incidence of several AEs. Conclusively, our meta-analysis suggests that the 2/1 schedule holds promise as an alternative means of reducing AEs and maintaining patient quality of life. While the survival outcomes of the 2/1 schedule seem also to be favorable, the level of evidence for this was low, and the interpretation of these findings should warrant caution. Large scale randomized trials are needed to support these results.

## 1. Introduction

The therapeutic concept of metastatic renal cell carcinoma (mRCC) has been dramatically changed through the development of targeted therapies, which result in significant improvement of clinical outcomes [1]. Sunitinib is an oral inhibitor of vascular endothelial growth factor (VEGF) receptors 1, 2, and 3 as well as platelet-derived growth factor receptors. Sunitinib is currently used as the first choice in the treatment of mRCC, and it has been shown to increase progression-free survival (PFS) when compared to interferon-α. The standard dosing schedule for sunitinib is 50 mg once daily for four weeks on and two weeks off (4/2). In the original phase III trial, 38% and 32% of patients taking sunitinib experienced dose interruptions and reductions, respectively, due to secondary toxicity [2]. Treatment-related adverse events (AEs) can make it difficult to maintain the standard dosing schedule of sunitinib, AEs may cause a decrease in quality of life (QOL) for these patients, as well as an increase in costs due to concomitant medication or conservative treatment.

Accordingly, some alternative schedules have been tested in an attempt to improve the safety profile and increase the dose intensity of sunitinib [3,4,5,6,7,8,9,10,11,12,13]. A continuous once daily dosing regimen with 37.5 mg of sunitinib has been proposed as an alternative [14], although the most often attempted strategy is a two weeks on and one week off (2/1) schedule. Several researchers have reported the clinical outcomes of the 2/1 schedule, including one phase II randomized controlled trial (RCT) [3,6,8,9,10,11,12,13,15], however, there has been no phase III RCT comparing 2/1 and 4/2 schedules of sunitinib. Recently, two meta-analysis studies have been published on this topic [16,17]. These two studies consistently suggested that the 2/1 dosing schedule was superior to the 4/2 dosing schedule in terms of treatment-related AEs; however, they showed conflicting results for survival outcomes. Therefore, in order to compare the efficacy and safety between the 2/1 and the 4/2 schedules, we performed an updated systematic review and meta-analysis, including all recently published studies. In addition, the quality of evidence was assessed using complementary statistical methods for assessing the certainty of the generated evidence.

## 2. Materials and Methods

### 2.1. Search Strategy

A literature search of all publications that appeared before August 2019 was performed using Embase (a biomedical literature database), PubMed, and Cochrane library. In addition, a cross-reference search of eligible articles was performed to check studies that were not found during the computerized search. We used combinations of the following MeSH terms and keywords: ‘sunitinib’, ‘renal cell’, ‘cancer’, ‘carcinoma’, ‘schedule’, ‘regimen’, and relevant variants. Conference abstracts were excluded even if they met eligibility criteria. Only trials published in English were included. The search produced 648 articles. Two authors (D.Y.C., J.W.K.) independently reviewed the titles and abstracts based on pre-established inclusion criteria and reviewed identified articles.

### 2.2. Inclusion Criteria and Study Eligibility

Eligibility of a study was evaluated using the consideration of participants, interventions, comparators, outcomes, and study design approach (PICOS), and the Preferred Reporting Items for Systematic Reviews and Meta-Analyses (PRISMA) guidelines. We defined the study population as patients diagnosed with mRCC, and the intervention as administration of sunitinib on a 2/1 schedule from the beginning of treatment, without any changes to the schedule. The comparator was defined as patients who had only undergone a 4/2 sunitinib schedule. The outcomes measured were AEs rate, PFS rate, and overall survival (OS) rate. The following inclusion criteria were used: (1) human subject research; (2) the baseline characteristics of patients from two groups must be comparable, including the total number of subjects and values of each index; (3) the comparator group must include patients with mRCC who had undergone treatment with a 4/2 sunitinib schedule; (4) the study group must contain patients with mRCC who had undergone treatment with a 2/1 sunitinib schedule; (5) outcome values reported must include rates of AEs, and/or rates of PFS and OS. In addition, both randomized and nonrandomized clinical trials were included.

### 2.3. Data Extraction

Two authors (D.Y.C., J.W.K.) independently reviewed the included articles and extracted the data at the trial level for each trial. Any discrepancies in the extracted data between the two reviewers were resolved through consensus. Extracted data included details on study design, inclusion and exclusion criteria, whether participants were randomized or nonrandomized, participant demographics and oncological characteristics, patient treatment characteristics (dosing schedule and median follow-up period), outcomes measured (PFS, OS, number of AEs, and hazard ratios (HR), 95% confidence intervals (CI), and *p*-values). The primary endpoint was the incidence of AEs. AEs were graded according to the National Cancer Institute—Common Terminology Criteria for Adverse Events, version 3.0 or 4.0. The incidences of overall and high-grade (grade 3 or higher) AEs were investigated separately. Secondary endpoints were the oncological outcomes, PFS and OS. Progression was defined as the progression of distant metastasis based on response evaluation criteria in solid tumors (RECIST) version 1.0 or 1.1 after initiation of sunitinib. OS was defined as the interval between the initiation of sunitinib and death due to mRCC or any other cause.

### 2.4. Study Quality Assessments

After the final group of articles was agreed upon, two authors (D.Y.C., J.W.K.) independently examined the quality using the Cochrane risk of bias tool and the Newcastle–Ottawa Scale (NOS). The Cochrane risk of bias tool for quality assessments of RCTs was recommended by the Cochrane Handbook for Systematic Reviews of Interventions [18]. The Cochrane risk of bias tool for RCTs is a seven-item list, which is designed to assess the following: (1) random sequence generation, (2) allocation concealment, (3) blinding of participants and personnel, (4) blinding of outcome assessment, (5) incomplete outcome data addressed, (6) selective reporting, and (7) other potential biases. Each item is further divided into three levels: high, unclear, and low risk of bias. Additionally, quality evaluation of included nonrandomized studies was performed according to the Newcastle–Ottawa Scale (NOS) [19]. The three major assessment categories of the NOS include selection, comparability, and exposure. A study can be given a rating of up to nine stars, and a final score of six stars or more is considered high quality. We assessed the quality of the generated evidence using the Grading of Recommendations, Assessments, Developments, and Evaluation (GRADE) system [20]. GRADE is used to systematically approach the evaluation and strength of recommendations. It consists of domains for methodology evaluation, accuracy of results, consistency of results, immediacy, and risk of publication bias. Based on these five criteria, the quality of evidence was rated as belonging to one of four levels (high, moderate, low, and very low).

### 2.5. Statistical Analysis

The effects of dosing schedules of sunitinib on oncological outcomes (PFS and OS) were measured using hazard ratios (HR). Log HR values were obtained directly from the papers reporting HR point estimates and CI, and the standard errors of log-HR were calculated using published CI [21]. Although some trials reported Kaplan–Meier log-rank *p*-values, they omitted HR or 95% CI, or both. In these cases, we estimated HR and 95% CI using *p*-values, number of total events, and number of participants that were randomized to each arm [22]. Also, for each study we assessed the risk ratio (RR) and the corresponding 95% CI of incidence of AEs. Pooled HR or RR with 95% CI indicated the effects according to dosing schedules on OS, PFS, and AEs. Estimates were then combined using a random effects model [23]. Chi square heterogeneity tests were used to test for statistical heterogeneity between trials. The *I^2^* statistic was calculated to measure discrepancies between clinical trials. A Cochran Q statistic *p*-value < 0.05 or I^2^ statistic > 50% was used to indicate statistically significant heterogeneity between trials [24]. If 10 studies or more that investigated a particular outcome were included, the use of funnel plots to assess small study effects was planned. However, fewer than 10 studies qualified for this review. A sensitivity analysis was performed by assessing the stability of results when each included study was sequentially excluded. Review Manager v.5.3 (2008; Nordic Cochrane Center, Cochrane Collaboration, Copenhagen, Denmark) was used for performing the meta-analysis. All *p*-values were two-sided, and except for the test of discrepancy, a *p*-value < 0.05 was considered statistically significant.

## 3. Results

### 3.1. Systematic Review Process

The results for the PRISMA flow diagram are presented in Figure 1. The initial database search found 648 studies (363 in PubMed, 242 in EMBASE, and 43 in the Cochrane library); among them, 242 studies remained after duplicates were removed. The titles and abstracts were then examined. As a result of this review, 210 articles were excluded. Subsequently, analysis of the full text articles was performed based on pre-established inclusion criteria. Finally, 11 studies with a total of 1012 patients were included. Information on the included studies is presented in Table 1. Both retrospective observational studies and one RCT were included. Of the 11 studies, 1 was an RCT and the remaining 10 were retrospective studies. The RCT was conducted in South Korea. Of the retrospective studies, four were conducted in Japan, two in China, one in USA, one in Egypt, one in Canada, and one was a multicenter study that included patients from several European countries. All ten trials enrolled patients diagnosed with mRCC who had undergone sunitinib treatment according to either 2/1 or 4/2 dosing schedules.

### 3.2. Quality Assessment

The results of the quality assessment based on the Cochrane risk of bias tool of the one included RCT are shown in Table 2A. In the RCT, there were two main sources of bias. The first was an unblinded study design (of open label nature), which could have led to bias because patients may switch from one arm to the other based on personal preferences. The second source of probable bias was that, in this study, the priority was selecting the schedule that had a higher failure-free survival rate at six months; therefore, other results (progression-free survival and overall survival) may have been influenced. Results of the quality assessment using NOS for the included nonrandomized studies are shown in Table 2B. Six out of the nine studies received a score of 6 points (indicating high quality). The investigation by Suo et al., Pan et al., and Zhang et al. received a score of 5 points due to insufficient information in the paper, which did not allow reviewers to determine whether the same method of ascertainment had been used in cases and controls.

### 3.3. Oncological Outcomes; Progression-Free Survival and Overall Survival

A total of seven nonrandomized studies [3,8,9,10,11,12,13] were included in a comparison of oncological outcomes according to dosing schedules. However, two studies [3,8] did not report an analysis of OS; therefore, OS was analyzed using data from the other five studies. One RCT by Lee at al. [6] was not included in the meta-analysis for survival outcomes due to the heterogeneity in study design. PFS and OS analysis were performed in two ways, using either unadjusted or adjusted HRs.

In the analysis using unadjusted HRs, meta-analysis revealed an overall HR of 0.65 for PFS in patients receiving the 2/1 dosing schedule (*p* = 0.0003; 95% CI, 0.51–0.82). Heterogeneity was found across studies (Cochran Q statistic, *p* = 0.03; I^2^ statistic, 56%), and an overall HR of 0.70 was revealed for OS in patients receiving the 2/1 dosing schedule (*p* = 0.01; 95% CI, 0.53–0.93). No heterogeneity across studies was found (Cochran Q statistic, *p* = 0.21; I^2^ statistic, 32%) (Figure 2).

In the analysis using adjusted HRs, meta-analysis revealed an overall HR of 0.58 for PFS in patients receiving the 2/1 dosing schedule (*p* = 0.005; 95% CI, 0.39–0.84). No heterogeneity was found across studies (Cochran Q statistic, *p* = 0.22; *I*^2^ statistic, 33%). Finally, it revealed an overall HR of 0.66 for OS in patients receiving the 2/1 dosing schedule (*p* = 0.08; 95% CI, 0.42–1.04). No heterogeneity across studies was found (Cochran Q statistic, *p* = 0.83; *I*^2^ statistic, 0%) (Figure 2).

The assessment of the quality of evidence of each comparison using the GRADE approach is shown in Table 3. Certainty was “very low” in all comparisons.

### 3.4. Incidence of Adverse Events

Nine studies [3,4,5,6,7,8,9,11,12] were included for assessment of AE incidence. Both overall and high-grade incidence of 15 AEs were investigated (Table 4). Fifteen AEs were classified into three categories; laboratory AEs (hypothyroidism, leukopenia, anemia, thrombocytopenia, and liver dysfunction), gastrointestinal AEs (anorexia, nausea, vomiting, diarrhea, dysgeusia), and others (hand–foot syndrome, hypertension, fatigue, stomatitis, skin color change). A meta-analysis was performed for the comparison of AE incidence according to dosing schedules. In the 2/1 sunitinib dosing schedule patients, there were statistically significant reductions in both overall and high-grade incidence of fatigue, hypertension, stomatitis, leukopenia, and skin color change. In addition, overall incidence of diarrhea, hand–foot syndrome, hypothyroidism, and dysgeusia significantly decreased in the 2/1 schedule patients. Finally, high-grade incidence of thrombocytopenia was significantly lower in patients receiving the 2/1 dosing schedule compared to those receiving the 4/2 schedule (Table 5 and Table 6).

## 4. Discussion

Our updated meta-analysis revealed that a sunitinib 2/1 schedule may improve oncological outcomes and safety profiles, compared to a conventional 4/2 schedule, in patients with mRCC. In terms of safety profiles, two meta-analysis [16,17] comparing the efficacy and safety between the sunitinib 2/1 and 4/2 schedules showed similar outcomes advocating the sunitinib 2/1 schedule, and our results are also consistent with their findings. However, the results for survival outcomes are conflicting. Chen et al. enrolled three studies for analyzing survival outcomes, and they concluded that there was no difference in PFS and OS between the 2/1 and 4/2 schedules [16]. However, Sun et al. included five studies for survival analysis, and among them, four studies were selected for PFS analysis, and another combination of four studies was used for OS analysis [17]. They indicated that both PFS and OS outcomes were significantly improved with the 2/1 schedule, compared to the 4/2 schedule. In our analysis, the 2/1 schedule showed significantly better PFS outcomes when using both unadjusted HRs from seven studies and adjusted HRs from four studies. OS outcomes were also better with the 2/1 schedule when applying unadjusted HRs from five studies, while there was no difference in OS outcomes between two schedules when using adjusted HRs from three studies.

These contradictory results were mainly derived from differences in the included studies, and incorporating more recently published studies, our study included more studies than previous meta-analyses. In addition, we excluded several studies included in previous meta-analyses, because those studies enrolled patients who underwent a schedule switch or another alternative schedules in the 2/1 schedule arm. Sun et al.’s meta-analysis included one phase II RCT in addition to other observational studies, although it is generally recommended that RCT and nonrandomized studies should not be combined in meta-analyses because it can result in increased heterogeneity [25]. Moreover, the previous studies used a mixture of unadjusted and adjusted results in the meta-analysis, while we pooled unadjusted and adjusted results separately. An unadjusted finding is the bivariate relationship between an independent and dependent variable that does not control for covariates or confounders. In cohort studies, unadjusted findings are sometimes presented, but they are generally recognized as having high potential for bias due to confounding. Therefore, for these types of studies, adjusted findings are usually preferred for meta-analysis [26]. In addition, we believe that this conflicting OS result may depend not only on statistical limitations, but also on the response to the drugs following sunitinib, such as axitinib, cabozantinib, and immune checkpoint inhibitors [27,28]. The currently included studies did not specify drug schedules after sunitinib, thus it is difficult to investigate adequate drug sequencing after sunitinib. Additional research on this topic is needed. We also provided the quality of evidence for the synthesized outcome by applying the GRADE approach, and the level of evidence for all comparisons was very low, owing to the nature of retrospectively designed studies and a small number patients—very low quality means very little confidence in the effect estimate, thus careful interpretation is required. Our meta-analysis demonstrated a positive benefit of the 2/1 dosing schedule in terms of various kinds of AEs. Of the 15 AEs analyzed, the overall or high-grade incidence of 10 AEs was significantly reduced when using the 2/1 schedule. In particular, the 2/1 dosing schedule was characterized by a decrease in fatigue, stomatitis, diarrhea, hand–foot syndrome, and dysgeusia, which directly affect the QOL of the patients [29]. Fatigue, especially, is known to have a significant impact on QOL [30]. In our results, both overall and high-grade fatigue decreased in patients treated with the 2/1 sunitinib dosing schedule. Therefore, this alternative schedule may provide a great advantage for patient QOL. In addition, in a study by Houk et al., side effects such as hypertension and hypothyroidism, which require the use of additional drugs and tests, were found to be decreased in patients on the 2/1 schedule [31]. This could lead to a reduction in medical costs for patients receiving sunitinib treatment. In addition, a reduction in hematologic side effects, such as thrombocytopenia and leukocytopenia, may decrease the need for dose reduction or discontinuation. Some studies have reported a higher incidence of AEs in patients receiving sunitinib when compared to the incidence of AEs in patients receiving both pazopanib and sunitinib, which are also first choices of treatment for mRCC [32,33]. In addition, several studies have reported that dose-adjustment according to the individual is effective [34,35,36]. Although the 2/1 schedule may be a way to maintain more drugs because of fewer AEs compared to the 4/2 schedule, it is important to adjust the concentration of the drug appropriately according to the condition of the patient rather than a uniform schedule.

Lee et al. conducted a phase II RCT comparing the efficacy and safety between the 2/1 and 4/2 schedules, and demonstrated that the sunitinib 2/1 schedule was associated with less toxicity and higher failure-free survival at six months than a 4/2 schedule, without compromising efficacy in terms of objective response rate and time to progression [6]. Another phase II clinical trial evaluating the stability of the 2/1 dosing schedule was recently published [15]. Although this study did not directly compare the results of the 2/1 schedule with the conventional 4/2 schedule, the oncological outcomes were similar to the outcomes of previously conducted conventional schedule studies. It also found a decrease in the incidence of some AEs, and a decrease in the number of patients requiring dose reduction. Jonasch et al. reported a decrease in dose reduction rates and a reduction in the incidence of AEs above grade 3 that favorably compared with findings from a phase II, COMPARZ study [15]. Although there has been no phase III RCT on this issue, the above-mentioned phase II clinical trials suggest that the 2/1 sunitinib schedule is associated with less toxicity than the 4/2 schedule without compromising oncological efficacy. In our meta-analysis, additional evidence was synthesized from nonrandomized studies, supporting the efficacy and safety of the 2/1 sunitinib schedule.

Our study had several limitations. First, the most included studies had retrospective designs that could not avoid inevitable limitations, such as selection bias. The small number of studies and sample size may affect the overall data quality. Second, the number of included studies in the present analysis was small. Finally, some of the studies in our review did not provide accurate HR and 95% CI for PFS and OS. Therefore, the estimates obtained using Kaplan–Meier curves were likely to have had errors. Also, about half of the studies did not provide adjusted HRs for PFS and OS throughout the multivariate analysis, so the number of studies was limited when the adjusted results were pooled for survival analysis. Therefore, well-designed RCTs should be conducted to overcome these limitations. Despite these limitations, our study has several strengths. Compared to previous meta-analyses, we updated the included studies with recently published literature, and extracted the data from the more strictly selected studies. In addition, we followed the general recommendations for meta-analysis from nonrandomized studies to avoid methodological flaws.

## 5. Conclusions

Our meta-analysis suggests that an alternative 2/1 sunitinib dosing schedule may have better PFS than the conventional 4/2 sunitinib schedule. However, its level of evidence was very low, so the interpretation of this result should be cautious. Moreover, the 2/1 schedule was beneficial for reducing the incidence of AEs. Accordingly, the 2/1 sunitinib dosing schedule holds promise as an alternative means of reducing AEs, maintaining patient QOL and prolonging treatment. We also believe that prospective large-scale studies of a 2/1 alternative schedule that demonstrate these advantages are needed.

## Figures and Tables

**Figure 1 cancers-11-01830-f001:**
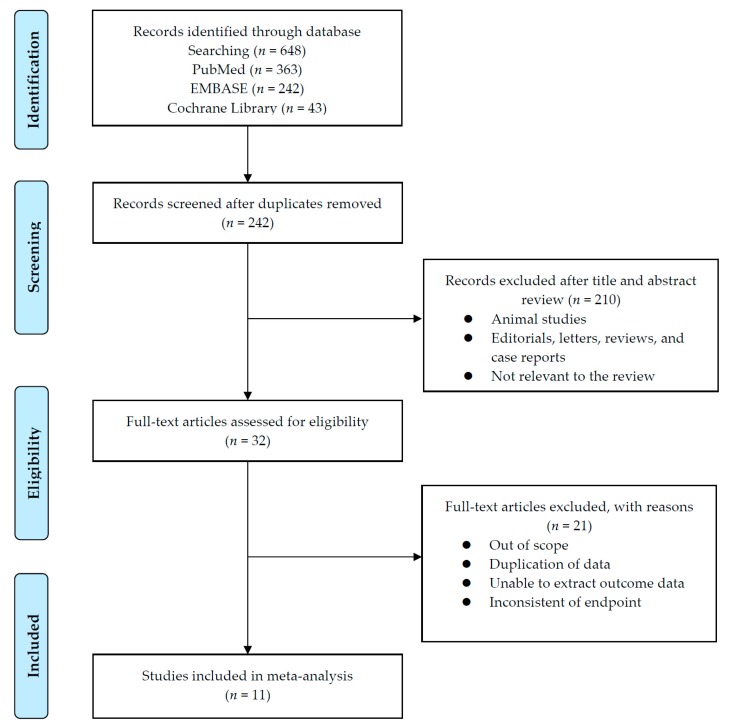
Flowchart of Preferred Reporting Items for Systematic Reviews and Meta-analysis (PRISMA).

**Figure 2 cancers-11-01830-f002:**
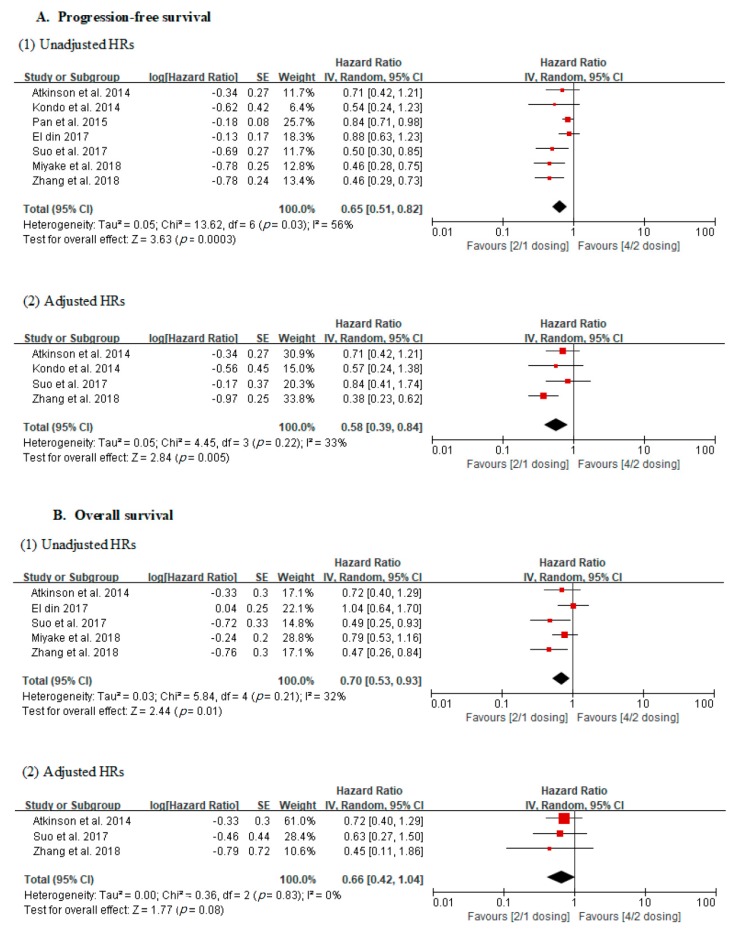
Forest plots of oncological outcomes according to dosing schedules.

**Table 1 cancers-11-01830-t001:** Characteristics of eligible studies.

Author(s) (Year)	Country	Study Design	Study Summary	Interval of Imaging Studies Tumor Response Evaluation Adverse Events Evaluation	Schedule	No. of Patients	Follow up (Months)	PFS (Months, IQR)	OS (Months, IQR)
Atkinson et al. (2014)	USA	Retrospective	Comparisons of oncological outcomes and incidence of adverse events between two groups: 2/1 and 4/2, 4/2 to 2/1 switch	NR RECIST v.1.1 NCI CTCAE v.4.0	2/1 dosing	24	NR	Median 11.6 (5.8–18.3)	Median 27.7 (21.2–NE)
					4/2 dosing	98		Median 4.3 (3.4–6.4)	Median 17.7 (10.8–22.2)
Kondo et al. (2014)	Japan	Retrospective	Comparisons of oncological outcomes and incidence of adverse events between two groups: 2/1 and 4/2	CT every 6–12 weeks RECIST v.1.1 NCI CTCAE v.4.0	2/1 dosing	26	Mean 11.9 ± 8.1 (range 2.6–31.6)	Median 18.4 (NR)	NR
					4/2 dosing	22	Mean 13.3 ± 10.1 (range 1.5–39.1)	Median 9.1 (NR)	
Najjar et al. (2014)	Japan	Retrospective	Comparisons of incidence of adverse events between two groups: 2/1 and 4/2	CT every 12 weeks RECIST v.1.1 NCI CTCAE v.4.0	2/1 dosing	30	Median 11.9 (range 0.9–73.3)	NR	NR
					4/2 dosing	30	Median 12.6 (range 1.2–62)		
Bracarda et al. (2015)	Europe	Retrospective	Comparisons of oncological outcomes and incidence of adverse events among three groups: 2/1, 4/2, and 4/2 to 2/1 switch	CT or MRI every 12 ± 1 weeks NR NCI CTCAE v.3.0 or v.4.0	2/1 dosing	41	Median 7.8 (IQR 5.8–22.4)	NR	NR
					4/2 dosing	208	Median 4.3 (IQR 2.0–12.0)		
Lee et al. (2015)	South Korea	Randomized controlled	Comparisons of oncological outcomes and incidence of adverse events between two groups: 2/1 and 4/2	CT every 12 weeks RECIST v.1.0 NCI CTCAE v.3.0 or v.4.0	2/1 dosing	38	Median 30.0 (IQR 19.5–53.6)	Median 12.1 (4.0–25.3)	Median 30.5 (18.9–42.0)
					4/2 dosing	36		Median 10.1 (7.5–12.7)	Median 28.4 (11.3–45.4)
Miyake et al. (2015)	Japan	Retrospective	Comparisons of incidence of adverse events between two groups: 2/1 and 4/2	CT every 12 weeks RECIST v.1.0 NCI CTCAE v.3.0	2/1 dosing	45	Median 3.4 (range 1.3–19.7)	NR	NR
					4/2 dosing	45	Median 8.9 (range 2.3–21.4)		
Pan et al. (2015)	China	Retrospective	Comparisons of oncological outcomes and incidence of adverse events among three groups: 2/1, 4/2, and 4/2 to 2/1 switch	NR RECIST v.1.1 NCI CTCAE v.4.0	2/1 dosing	32	Median 1.5 years (range 1.0–2.2)	Median 11.2 (NR)	NR
					4/2 dosing	50	Median 1.9 years (range 1.3–2.7)	Median 9.5 (NR)	NR
El Din (2017)	Egypt	Retrospective	Comparisons of oncological outcomes and incidence of adverse events between two groups: 2/1 and 4/2	CT or MRI every 18 weeks NR NCI CTCAE v.4.0	2/1 dosing	26	Median 23 (range 3–43)	Median 17 (2–43)	Median 24 (2–42)
					4/2 dosing	30	Median 24 (range 2–42)	Median 15 (1–42)	Median 23 (3–43)
Suo et al. (2017)	Canada	Retrospective	Comparisons of oncological outcomes and incidence of adverse events among three groups: 2/1, 4/2, 2/1 modified dosing (37.5 mg, 25 mg, or 12.5 mg) and continuous dosing (37.5 mg, 25 mg, or 12.5 mg)	NR RECIST NR	2/1 dosing	9	NR	Median 6.0 (NR)	Median 23.1 (NR)
					4/2 dosing	59		Median 3.0 (NR)	Median 11.2 (NR)
Miyake et al. (2018)	Japan	Retrospective	Comparisons of oncological outcomes and incidence of adverse events among three groups: 2/1, 4/2, and 4/2 to 2/1 switch	CT every 6–12 weeks RECIST v.1.1 NCI CTCAE v.3.0	2/1 dosing	47	NR	Median 13.8 (NR)	Median 39.2 (NR)
					4/2 dosing	62		Median 6.3 (NR)	Median 30.8 (NR)
Zhang et al. (2018)	China	Retrospective	Comparisons of oncological outcomes and incidence of adverse events among three groups: 2/1, 4/2, and 4/2 to 2/1 switch	NR RECIST v.1.1 NCI CTCAE v.4.0	2/1 dosing	24	Median 37	Median 11 (NR)	Median 28 (NR)
					4/2 dosing	30		Median 12.5 (NR)	Median 21 (NR)

CT, computed tomography; IQR, interquartile range; MRI, magnetic resonance imaging; NE, not estimable; NR, not reported; NCI CTCAE, National Cancer Institute Common Terminology Criteria for Adverse Events version; OS, overall survival; PFS, progression free survival; RECIST, The Response Evaluation Criteria in Solid Tumor; US, ultrasonography.

**Table 2 cancers-11-01830-t002:** Results of quality assessment by the Cochrane risk of bias tool (A) and Newcastle–Ottawa Scale (B).

**A. Quality assessment of a randomized controlled trial**											
**Author(s) (Year)**	**Random Sequence Generation (Selection Bias)**		**Allocation Concealment (Selection Bias)**		**Blinding of Participants and Personnel (Performance Bias)**	**Blinding of Outcome Assessment (Detection Bias)**	**Incomplete Outcome Data Addressed (Attrition Bias)**		**Selective Reporting (Reporting Bias)**		**Other Bias**
Lee et al. (2015)	Low risk		Low risk		High risk	High risk	Low risk		Low risk		Unclear
**B. Quality assessment of nonrandomized studies**											
**Author(s) (Year)**	**Selection (4)**						**Comparability (2)**	**Exposure (3)**			**Total score**
	**Adequate Definition of Cases**	**Representativeness of Cases**		**Selection of Controls**		**Definition of Controls**	**Control for Important Factor or Additional Factor**	**Ascertainment of Exposure**	**Same Method of Ascertainment for Cases and Controls**	**Non-Response Rate**	
Atkinson et al. (2014)	1	1		0		0	2	1	1	0	6
Kondo et al. (2014)	1	1		0		0	2	1	1	0	6
Najjar et al. (2014)	1	1		0		0	2	1	1	0	6
Bracarda et al. (2015)	1	1		0		0	2	1	1	0	6
Miyake et al. (2015)	1	1		0		0	2	1	1	0	6
Pan et al. (2015)	1	1		0		0	2	1	0	0	5
El Din (2017)	1	1		0		0	2	1	1	0	6
Suo et al. (2017)	1	1		0		0	2	1	0	0	5
Miyake et al. (2018)	1	1		0		0	2	1	1	0	6
Zhang et al. (2018)	1	1		0		0	2	1	0	0	5

**Table 3 cancers-11-01830-t003:** Results of the GRADE (Grading of Recommendations, Assessments, Developments, and Evaluation) quality assessment of direct evidence of each comparison.

No. of Studies	Study Design	Risk of Bios	Inconsistency	Indirectness	Imprecision	Other Consideration	No. of Patients		Effect	Overall Quality of Evidence
							2/1 Schedule	4/2 Schedule		
1. Progression-free survival										
7 Unadjusted	observational studies	not serious	not serious	not serious	Serious *	none	188	351	HR 0.66 (0.54–0.82)	Very low
4 Adjusted	observational studies	not serious	not serious	not serious	Serious **	none	83	209	HR 0.58 (0.39–0.84)	Very low
2. Overall survival										
5 Unadjusted	observational studies	not serious	not serious	not serious	Serious *	none	130	279	HR 0.75 (0.57–0.99)	Very low
3 Adjusted	observational studies	not serious	not serious	not serious	Serious **	none	57	187	HR 0.66 (0.42–1.04)	Very low

*: Apply to unadjusted values; **: Total number of participants is small.

**Table 4 cancers-11-01830-t004:** Summary of adverse events investigated in the current study.

Study (year)	Schedule	No. of Patients	Complication (No.)																													
			Hypo-Thyroidism		Leukopenia		Anemia		Thrombo-cytopenia		Liver Dysfunction		Anorexia		Nausea		Vomiting		Diarrhea		Dysgeusia		HFS		HTN		Fatigue		Stomatitis		Skin Color Change	
			Al *	HG	Al *	HG *	All	HG	All	HG *	All	HG	All	HG	All	HG	All	HG	All *	HG	All *	HG	All *	HG	All *	HG *	All *	HG *	All *	HG *	All *	HG
Kondo et al. (2014)	2/1	26	10	0	22	5	20	3	24	5	18	0	11	1	NR	NR	NR	NR	9	0	6	0	15	0	16	0	19	1	10	0	NR	NR
	4/2	22	13	0	18	1	19	1	19	6	16	0	14	1	NR	NR	NR	NR	16	1	10	0	19	2	17	0	19	1	13	0	NR	NR
Najjar et al. (2014)	2/1	30	13	0	1	1	NR	NR	2	0	NR	NR	2	1	2	1	NR	NR	11	1	NR	NR	5	0	8	2	16	2	3	0	NR	NR
	4/2	30	11	2	6	3	NR	NR	6	3	NR	NR	7	2	6	2	NR	NR	12	6	NR	NR	15	8	8	6	21	11	6	2	NR	NR
Bracarda et al. (2015)	2/1	41	11	1	NR	NR	NR	NR	10	0	NR	NR	6	2	8	1	1	0	2	5	7	0	15	2	8	1	26	2	14	1	NR	NR
	4/2	208	77	3	NR	NR	NR	NR	69	16	NR	NR	54	5	63	6	18	1	87	8	68	1	116	21	95	19	155	21	127	14	NR	NR
Lee et al. (2015)	2/1	38	17	0	14	4	27	5	27	9	9	0	21	0	12	0	5	0	14	0	NR	NR	26	7	21	9	22	1	27	1	13	0
	4/2	36	13	0	22	10	26	3	28	8	11	0	18	0	9	0	6	0	5	1	NR	NR	27	13	26	12	30	2	31	4	20	0
Miyake et al. (2015)	2/1	45	20	0	33	3	26	3	41	13	NR	NR	NR	NR	NR	NR	NR	NR	16	0	10	0	15	1	16	1	13	4	11	0	19	0
	4/2	45	28	1	36	8	28	4	44	23	NR	NR	NR	NR	NR	NR	NR	NR	27	1	13	0	25	5	25	5	23	8	13	0	26	0
Pan et al. (2015)	2/1	32	19	1	8	3	3	1	8	2	NR	NR	NR	NR	NR	NR	NR	NR	10	2	NR	NR	15	2	16	1	16	1	9	1	NR	NR
	4/2	50	28	2	33	8	15	2	18	7	NR	NR	NR	NR	NR	NR	NR	NR	32	7	NR	NR	42	5	25	4	43	5	23	1	NR	NR
El Din (2017)	2/1	26	9	1	10	1	11	1	4	0	2	0	NR	NR	Nausea and vomiting combine number				3	NR	NR	NR	9	0	6	1	10	1	2	1	3	NR
	4/2	30	12	4	15	1	11	3	13	3	3	2	NR	NR	NR	NR	NR	NR	11	NR	NR	NR	21	6	15	4	21	8	11	6	4	NR
Miyake et al. (2018)	2/1	47	23	0	34	4	25	5	43	4	20	0	NR	NR	NR	NR	NR	NR	20	0	NR	NR	21	5	17	2	14	5	NR	NR	20	0
	4/2	62	40	2	50	7	42	6	61	17	34	0	NR	NR	NR	NR	NR	NR	40	2	NR	NR	34	6	37	5	42	11	NR	NR	38	0
Zhang et al. (2018)	2/1	24	9	0	11	2	6	1	8	2	8	2	5	0	NR	NR	NR	NR	7	1	NR	NR	10	5	4	2	10	3	7	1	4	0
	4/2	30	17	2	17	9	16	2	16	2	8	1	12	1	NR	NR	NR	NR	11	4	NR	NR	15	3	12	4	17	7	11	4	6	1

* Statistically significant value; Reference value; 4/2 dosing. HFS, hand–foot syndrome; HG, high-grade; HTN, hypertension; NR, not reported.

**Table 5 cancers-11-01830-t005:** Meta-analysis of all grade adverse events according to dosing schedules.

Adverse Events	No. Studies	Dosing Schedule	No. of Patients	RR	*p*-Value	*I*^2^ (%)	*p*_H_-Value
				(95% CI)			
Laboratory abnormalities							
Hypothyroidism *	9	2/1	309	0.84 (0.72–0.99)	0.04	0	0.53
		4/2	513				
Leukopenia *	8	2/1	268	0.79 (0.63–0.99)	0.04	60	0.01
		4/2	205				
Anemia	7	2/1	238	0.86 (0.72–1.03)	0.10	27	0.22
		4/2	275				
Thrombocytopenia	9	2/1	309	0.89 (0.77–1.03)	0.11	62	0.007
		4/2	513				
Liver dysfunction	5	2/1	161	0.88 (0.70–1.12)	0.31	0	0.84
		4/2	180				
Gastrointestinal adverse events							
Anorexia	5	2/1	159	0.70 (0.47–1.04)	0.08	38	0.17
		4/2	326				
Nausea	3	2/1	109	0.77 (0.42–1.45)	0.41	38	0.2
		4/2	274				
Vomiting	2	2/1	79	0.62 (0.24–1.62)	0.33	0	0
		4/2	244				
Diarrhea *	8	2/1	309	0.62 (0.44–0.89)	0.010	62	0.007
		4/2	513				
Dysgeusia *	3	2/1	112	0.6 (0.39–0.92)	0.02	0	0.68
		4/2	275				
Other adverse events							
Hand–foot syndrome *	9	2/1	309	0.68 (0.58–0.81)	<0.00001	25	0.22
		4/2	513				
Hypertension *	9	2/1	309	0.70 (0.58–0.84)	0.0002	16	0.30
		4/2	513				
Fatigue *	9	2/1	309	0.69 (0.60–0.81)	<0.00001	29	0.19
		4/2	513				
Stomatitis *	8	2/1	262	0.70 (0.57–0.86)	0.0006	10	0.35
		4/2	451				
Skin color change *	4	2/1	180	0.70 (0.55–0.89)	0.004	0	0.98
		4/2	203				

* Statistically significant value; Reference value; 4/2 dosing, N.A, not available; RR, risk ratio.

**Table 6 cancers-11-01830-t006:** Meta-analysis of high-grade adverse events according to dosing schedules.

Adverse Events	No. Studies	Dosing Schedule	No. of Patients	RR	*p*-Value	I^2^ (%)	*p*_H_-Value
				(95% CI)			
Laboratory abnormalities							
Hypothyroidism	9	2/1	309	0.46 (0.17–1.23)	0.12	0	0.88
		4/2	513				
Leukopenia *	8	2/1	268	0.53 (0.32–0.87)	0.01	0	0.52
		4/2	305				
Anemia	7	2/1	238	1.02 (0.55–1.90)	0.95	0	0.90
		4/2	275				
Thrombocytopenia *	9	2/1	309	0.58 (0.40–0.83)	0.003	0	0.52
		4/2	513				
Liver dysfunction	5	2/1	161	0.91 (0.09–9.42)	0.94	35	0.21
		4/2	180				
Gastrointestinal adverse events							
Anorexia	5	2/1	159	1.05 (0.34–3.19)	0.93	0	0.7
		4/2	326				
Nausea	3	2/1	109	0.67 (0.14–1.45)	0.41	38	0.2
		4/2	274				
Vomiting	2	2/1	79	1.66 (0.07–40.02)	0.76	N.A	
		4/2	244				
Diarrhea	8	2/1	283	0.52 (0.19–1.41)	0.20	42	0.10
		4/2	483				
Dysgeusia	3	2/1	112	1.66 (0.07.40.02)	0.76	N.A	
		4/2	275				
Other adverse events							
Hand–foot syndrome	9	2/1	309	0.55 (0.29–1.02)	0.06	30	0.18
		4/2	513				
Hypertension *	9	2/1	309	0.51 (0.31–0.83)	0.008	0	0.90
		4/2	513				
Fatigue *	9	2/1	309	0.43 (0.26–0.70)	0.0007	0	0.89
		4/2	513				
Stomatitis *	8	2/1	262	0.32 (0.13–0.81)	0.02	0	0.89
		4/2	451				
Skin color change	4	2/1	154	0.41 (0.02–9.71)	0.58	N.A	
		4/2	173				

* Statistically significant value; Reference value; 4/2 dosing. N.A, Not available; RR, Risk ratio.

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
