# Peer review of "Does an Alternative Sunitinib Dosing Schedule Really Improve Survival Outcomes Over a Conventional Dosing Schedule in Patients with Metastatic Renal Cell Carcinoma? An Updated Systematic Review and Meta-Analysis"

_cancers, 2019, doi:10.3390/cancers11121830_

Round 1
Reviewer 1 Report
In the updated systematic review and metaanalysis by Chung and colleagues evaluate the influence of a modified dosing schedule of sunitinib (2/1 weeks) compared to the standard schedule (4/2 weeks) on OS, PFS and toxicity profile. Therefore comprehensive data analyses have been carried out and statistically evaluated. In addition, the quality of the trials is carefully apportioned and discussed for final conclusions.
The different dosing schedules are of high interest for physicians involved in renal cell carcinoma treatment. In fact, the 2/1 schedule is already widely used in clinical practice. Thus, the results of this metaanalysis are of significant relevance for clinical practice although most studies included were retrospective and only small patient numbers could been included. The authors critically discuss these limitations and incorporate the deficiencies in their conclusions.
The manuscript is well written and to the best of my knowledge statistical analyses were appropriately carried out.
Minor comments: There are few spelling mistakes especially in the dosing schedules which need to be corrected e.g. page 19 line 6 "4/1" needs to be changed to "4/2"
Author Response
Dear Reviewer
Thank you for your thoroughly reviewing our manuscript (cancers-619121) entitled “Does an alternative sunitinib dosing schedule really improve survival outcomes over a conventional dosing schedule in patients with metastatic renal cell carcinoma? An updated systematic review and meta-analysis” Also, we are grateful for the chance to revise our manuscript. Our manuscript has been carefully revised according to the reviewers’ comments. Please find our responses to the reviewer’ comments beginning on the next page.
We hope that our revised paper is acceptable for publication in Cancers, and we look forward to receiving your final decision.
Thanks, again.
Sincerely,
Kang Su Cho, M.D., Ph.D.
Department of Urology, Gangnam Severance Hospital, Yonsei University College
Reviewer #1’s comments
[Comment]
1) In the updated systematic review and metaanalysis by Chung and colleagues evaluate the influence of a modified dosing schedule of sunitinib (2/1 weeks) compared to the standard schedule (4/2 weeks) on OS, PFS and toxicity profile. Therefore comprehensive data analyses have been carried out and statistically evaluated. In addition, the quality of the trials is carefully apportioned and discussed for final conclusions.
The different dosing schedules are of high interest for physicians involved in renal cell carcinoma treatment. In fact, the 2/1 schedule is already widely used in clinical practice. Thus, the results of this metaanalysis are of significant relevance for clinical practice although most studies included were retrospective and only small patient numbers could been included. The authors critically discuss these limitations and incorporate the deficiencies in their conclusions.
The manuscript is well written and to the best of my knowledge statistical analyses were appropriately carried out.
[Answers]
Thank you for your comment. We also agree with your feedback. The studies included in our meta-analysis were retrospective and low in number, thus the level of evidence was very low according to the GRADE. This is a limitation of our research and we think that readers should be careful for the interpretation in our research. In result, we also believe that prospective large-scale studies of a 2/1 alternative schedule that demonstrate these advantages are needed. We have added these comments in discussion section and conclusion section as shown in below.
On page #19
Our study has several limitations. First, the most included studies had retrospective designs that could not avoid inevitable limitations, such as selection bias. The small number of studies and sample size may affect the overall data quality. Second, the number of included studies in the present analysis was small.
However, its level of evidence was very low, the interpretation of this result should be cautious.
We also believe that prospective large-scale studies of a 2/1 alternative schedule that demonstrate these advantages are needed.
[Comment]
2) Minor comments: There are few spelling mistakes especially in the dosing schedules which need to be corrected e.g. page 19 line 6 "4/1" needs to be changed to "4/2"
[Answers]
Thank you for your comment. Sorry for not correcting typo. We have modified the spelling according to your feedback.

Reviewer 2 Report
Chu et al. performed meta-analysis of all published studies on the alternative schedules of sunitinib in patients with metastatic renal cell carcinoma. Their analysis showed that a 2-weeks-on and 1-week-off (2/1) schedule was beneficial for reducing the incidence of several adverse effects (AEs) compared to a conventional schedule, but no difference in overall survival was observed.
Criticisms
The results of this study clarified the usefulness of the alternative schedule of sunitinib in patients with metastatic renal cell carcinoma. However, there are similar studies, and the difference between these studies and this manuscript is unclear. Furthermore, when the patients becomes resistant to sunitinib treatment, other drugs such as immune checkpoint inhibitors are often administered. Therefore, it is considered that there is not much prognostic significance by administering sunitinib alone. I think this paper will be more meaningful if the authors describe the significance of this treatment schedule in relation to recent advances in treatment of renal cell carcinoma patients.
Author Response
Dear Reviewer
Thank you for your thoroughly reviewing our manuscript (cancers-619121) entitled “Does an alternative sunitinib dosing schedule really improve survival outcomes over a conventional dosing schedule in patients with metastatic renal cell carcinoma? An updated systematic review and meta-analysis” Also, we are grateful for the chance to revise our manuscript. Our manuscript has been carefully revised according to the reviewers’ comments. Please find our responses to the reviewer’ comments beginning on the next page.
We hope that our revised paper is acceptable for publication in Cancers, and we look forward to receiving your final decision.
Thanks, again.
Sincerely,
Kang Su Cho, M.D., Ph.D.
Reviewer #2’s comments
[Comment]
Chu et al. performed meta-analysis of all published studies on the alternative schedules of sunitinib in patients with metastatic renal cell carcinoma. Their analysis showed that a 2-weeks-on and 1-week-off (2/1) schedule was beneficial for reducing the incidence of several adverse effects (AEs) compared to a conventional schedule, but no difference in overall survival was observed.
Criticisms
The results of this study clarified the usefulness of the alternative schedule of sunitinib in patients with metastatic renal cell carcinoma. However, there are similar studies, and the difference between these studies and this manuscript is unclear.
[Answers]
Thank you for your comment.
Compared to previous meta-analysis, we updated the included studies with recently published literatures, and extracted the data from the more strictly selected studies. The previous studies used a mixture of unadjusted and adjusted results in the meta-analysis, while we pooled unadjusted and adjusted results separately. An unadjusted finding is the bivariate relationship between an independent and dependent variable that does not control for covariates or confounders. In cohort studies, unadjusted findings are sometimes presented, but they are generally recognized as having high potential for bias due to confounding. Therefore, for these types of studies, adjusted findings are usually preferred for meta-analysis. In addition, it is generally recommended that RCT and nonrandomized studies should not be combined in meta-analyses because it can result in increased heterogeneity. We followed the general recommendations for meta-analysis from nonrandomized studies to avoid methodological flaws. We described it in the discussion section as shown below.
On page #18
Sun et al.’s meta-analysis included one phase II RCT in addition to other observational studies, although it is generally recommended that RCT and nonrandomized studies should not be combined in meta-analyses because it can result in increased heterogeneity [25]. Moreover, the previous studies used a mixture of unadjusted and adjusted results in the meta-analysis, while we pooled unadjusted and adjusted results separately. An unadjusted finding is the bivariate relationship between an independent and dependent variable that does not control for covariates or confounders. In cohort studies, unadjusted findings are sometimes presented, but they are generally recognized as having high potential for bias due to confounding. Therefore, for these types of studies, adjusted findings are usually preferred for meta-analysis [26].
On page #18
We also provided the quality of evidence for the synthesized outcome by applying the GRADE approach, and the level of evidence for all comparisons was “very low,” owing to the nature of retrospectively designed studies and small number patients: very low quality means very little confidence in the effect estimate, thus careful interpretation is required.
On page #19
Compared to previous meta-analysis, we updated the included studies with recently published literatures, and extracted the data from the more strictly selected studies. In addition, we followed the general recommendations for meta-analysis from nonrandomized studies to avoid methodological flaws.
[Comment]
Furthermore, when the patients becomes resistant to sunitinib treatment, other drugs such as immune checkpoint inhibitors are often administered. Therefore, it is considered that there is not much prognostic significance by administering sunitinib alone. I think this paper will be more meaningful if the authors describe the significance of this treatment schedule in relation to recent advances in treatment of renal cell carcinoma patients.
[Answers]
We also agree that the effect of treatment after sunitinib resistance or intolerability is very important especially on overall survival. Unfortunately, the included studies do not specify drug schedules after sunitinib, thus it is difficult to investigate the adequate drug sequencing after sunitinib. Additional researches on this topic are needed. According to your suggestions, we revised the discussion section as shown below.
On page #18
In addition, we believe that this conflicting OS result may depend not only on statistical limitations, but also on the response to the drugs following sunitinib such as axitinib, cabozantinib, and immune checkpoint inhibitors [27,28]. Currently included studies do not specify drug schedules after sunitinib, thus it is difficult to investigate the adequate drug sequencing after sunitinib. Additional researches on this topic are needed.

Reviewer 3 Report
Tha manuscript is well written, however I have concerns regarding the additional information added in the literature with this paper respect the two previous metanalysis.
Author Response
Dear reviewer
Thank you for your thoroughly reviewing our manuscript (cancers-619121) entitled “Does an alternative sunitinib dosing schedule really improve survival outcomes over a conventional dosing schedule in patients with metastatic renal cell carcinoma? An updated systematic review and meta-analysis” Also, we are grateful for the chance to revise our manuscript. Our manuscript has been carefully revised according to the reviewers’ comments.
We hope that our revised paper is acceptable for publication in Cancers, and we look forward to receiving your final decision.
Thanks, again.
Sincerely,
Kang Su Cho, M.D., Ph.D.
Reviewer 4 Report
This is an well done systemic review and provide useful clinical information.
Author Response

(The authors gave the same response as above.)

Reviewer 5 Report
This is a carefully done systemic analysis of the data for the sunitinib 2/1schedule. This schedule its commonly used to replace the rigid 4/2 schedule. The problem is that the 2/1 is just another rigid schedule that does not work for all patients either. Less toxicity can actually translate into less activity if patients get inadequate drug exposure. The authors should refer to and discuss two recent studies that use individualized TKI dose and schedule in an attempt to optimize outcome for all patients. Both studies use toxicity as a surrogate for adequate drug exposure. Both studies show that one dose or schedule do not work for all.
Bjarnason, G. A., et al. (2019). "The efficacy and safety of sunitinib given on an individualised schedule as first-line therapy for metastatic renal cell carcinoma: A phase 2 clinical trial." European Journal of Cancer 108: 69-77.
Ornstein, M. C., et al. (2019). "Individualised axitinib regimen for patients with metastatic renal cell carcinoma after treatment with checkpoint inhibitors: a multicentre, single-arm, phase 2 study." Lancet Oncol 20(10): 1386-1394.
The observation that patient that experience a degree of toxicity do better has been further supported by a recent Post Hock analysis of the COMPARZ trial showing improved ORR, PFS and OS in patients with a degree of toxicity leading to dose and schedule changes.
Sternberg, C. N., et al. (2019). "COMPARZ Post Hoc Analysis: Characterizing Pazopanib Responders With Advanced Renal Cell Carcinoma." Clin Genitourin Cancer. 10.1016/j.clgc.2019.01.015
Author Response
Dear Reviewer
Thank you for your thoroughly reviewing our manuscript (cancers-619121) entitled “Does an alternative sunitinib dosing schedule really improve survival outcomes over a conventional dosing schedule in patients with metastatic renal cell carcinoma? An updated systematic review and meta-analysis” Also, we are grateful for the chance to revise our manuscript. Our manuscript has been carefully revised according to the reviewers’ comments. Please find our responses to the reviewer’ comments beginning on the next page.
We hope that our revised paper is acceptable for publication in Cancers, and we look forward to receiving your final decision.
Thanks, again.
Sincerely,
Kang Su Cho, M.D., Ph.D.
Reviewer #5’s comment
[Comment]
This is a carefully done systemic analysis of the data for the sunitinib 2/1schedule. This schedule its commonly used to replace the rigid 4/2 schedule. The problem is that the 2/1 is just another rigid schedule that does not work for all patients either. Less toxicity can actually translate into less activity if patients get inadequate drug exposure. The authors should refer to and discuss two recent studies that use individualized TKI dose and schedule in an attempt to optimize outcome for all patients. Both studies use toxicity as a surrogate for adequate drug exposure. Both studies show that one dose or schedule do not work for all.
Bjarnason, G. A., et al. (2019). "The efficacy and safety of sunitinib given on an individualised schedule as first-line therapy for metastatic renal cell carcinoma: A phase 2 clinical trial." European Journal of Cancer 108: 69-77.
Ornstein, M. C., et al. (2019). "Individualised axitinib regimen for patients with metastatic renal cell carcinoma after treatment with checkpoint inhibitors: a multicentre, single-arm, phase 2 study." Lancet Oncol 20(10): 1386-1394.
The observation that patient that experience a degree of toxicity do better has been further supported by a recent Post Hock analysis of the COMPARZ trial showing improved ORR, PFS and OS in patients with a degree of toxicity leading to dose and schedule changes.
Sternberg, C. N., et al. (2019). "COMPARZ Post Hoc Analysis: Characterizing Pazopanib Responders With Advanced Renal Cell Carcinoma." Clin Genitourin Cancer. 10.1016/j.clgc.2019.01.015
[Answers]
Thank you for your helpful comment. As you mentioned, less toxicity can actually translate into less activity due to inadequate drug exposure, and one dose or schedule do not work for all. Although the 2/1 schedule may be a way to maintain more drugs because of less AEs compared to the 4/2 schedule, it is important to adjust the concentration of the drug appropriately according to the condition of the patient rather than the uniform schedule.
On page #18 & 19
In addition, several studies have reported that dose-adjustment according to individual is effective [34-36]. Although the 2/1 schedule may be a way to maintain more drugs because of less AEs compared to the 4/2 schedule, it is important to adjust the concentration of the drug appropriately according to the condition of the patient rather than uniform schedule.
